# Reanalysis of a Randomized Controlled Trial on Promoting Influenza Vaccination in General Practice Waiting Rooms: A Zelen Design

**DOI:** 10.3390/vaccines10050826

**Published:** 2022-05-23

**Authors:** Christophe Berkhout, Jeroen De Man, Claire Collins, Amy Willefert-Bouche, Suzanna Zgorska-Maynard Moussa, Margot Badelon, Lieve Peremans, Paul Van Royen

**Affiliations:** 1UFR3S, Department of General Practice/Family Medicine, Lille University, 59045 Lille, France; amywillefert@msn.com (A.W.-B.); moussa.suz@hotmail.com (S.Z.-M.M.); m.badelon@gmail.com (M.B.); 2Department of Family Medicine and Population Health, University of Antwerp, 2610 Antwerp, Belgium; jeroen.deman@uantwerpen.be (J.D.M.); lieve.peremans@uantwerpen.be (L.P.); paul.vanroyen@uantwerpen.be (P.V.R.); 3Irish College of General Practitioners, D02 XR68 Dublin, Ireland; claire.collins@icgp.ie; 4Department of Nursing and Midwifery, University of Antwerp, 2610 Antwerp, Belgium

**Keywords:** vaccination coverage, influenza, human, primary health care, health promotion, randomized controlled trials as topic

## Abstract

In 2014–2015, we conducted a randomized controlled trial (RCT) assessing the effect of an advertising campaign for influenza vaccination using posters and pamphlets in general practitioner (GP) waiting rooms. No effect of the intervention could be demonstrated, but the immunization uptake increased in both arms of the study. In 2019, we deepened the investigations explaining the increased uptake conducting a registry-based 4/2/1 cluster RCT designed by Zelen with two extra years of follow-up of the study cohort. The study population included 23,024 patients eligible to be vaccinated who were registered with 175 GPs. The main outcome remained the number of vaccination units delivered per study group. Data were extracted from the SNIIRAM warehouse claim database for the Lille-Douai district (northern France). No difference in vaccination uptake was found in the Zelen versus the control group of the initial RCT. Overall, the proportion of vaccinated patients increased in the cohort from 51.4% to 70.4% over the three years. Being vaccinated the previous year was a strong predictor of being vaccinated in a subsequent year. The increase in vaccination uptake, especially among people older than 65, can be explained by a cohort effect. Health promotion and the promotion of primary health care may play an important role in this increase.

## 1. Introduction

Seasonal influenza epidemics happen yearly in France and occur generally during 5 to 13 winter weeks (e.g., in 2016–2017, from week 49/2016 to week 06/2017) and can start early in December (e.g., for 2016–2017) or start late in February (e.g., for 2014–2015). Its intensity is variable with weekly peaks floating between 410 (2013–2014) and 990 (2014–2015) occurrences per 100,000 inhabitants. The mortality excess during the seasonal epidemic was 18,300 in 2014–2015 and 21,200 in 2016–2017, mainly in persons above the age of 75 [1].

Every year, persons 65 years and older and patients with a targeted, chronic condition (e.g., COPD, asthma, and diabetes) or pregnant women receive in October from their mandatory insurance company an invitation letter to undergo seasonal influenza vaccination, including a voucher to receive a free vaccination unit delivered by their community pharmacy between October and January. If patients lose their voucher for a free vaccination or did not receive it for their chronic condition or pregnancy, their GP can deliver a new one, but we found that the occurrence of this contingency was negligible [2]. At the same time, a nationwide vaccination campaign is broadcasted on TV and radio stations, in journals and newspapers, and in the offices of health professionals through posters and pamphlets in waiting rooms. Vaccination injections can be given by general practitioners (GPs), pharmacists, nurses, and midwives. The national vaccination uptake percentage of 45.7% in 2014–2015 placed France in the upper range of the European mean coverage of the target population [3,4].

Unless they are affected by a chronic condition, infants and children are not targeted for seasonal influenza vaccination in France.

During winter 2014–2015, we conducted a randomized controlled trial (RCT) to assess the effect of the annual influenza vaccination campaign in GP waiting rooms [2]. This trial compared the delivery of vaccines, counted as the number of vaccination units, in community pharmacies to patients of GPs whose waiting rooms displayed a poster and pamphlets promoting vaccination (intervention) and to patients of other GPs where the waiting rooms was as per their usual state (control group). Routinely collected data from the “*Système Informationnel de l’Assurance Maladie-Erasme*” (SIAM-ERASME), the main mandatory health insurance company covering about 75% of the population of the urban area of Lille-Douai in northern France, were used to compare both groups [5]. No difference was found between these groups, strengthening the evidence that exposure to posters and pamphlets in GP waiting rooms does not result in different health behaviors [6].

However, two interesting outcomes emerged: (1) In the case of previous vaccination in 2013–2014, the vaccination uptake probability in 2014–2015 was 5.63 times higher (95% confidence interval [5.21; 6.10]), and (2) the vaccination uptake increased by 3% in both arms of the research group from 46% [45.23–47.13] to 49% [48.04–49.95]. Contemporaneously, according to national public health data, the seasonal influenza vaccination uptake in this area decreased by 2.4% from 52.7% to 50.3% and on a national level by 2.8% from 48.9% to 46.1% [7]. Indeed, the vaccination uptake gradually decreased every year from 2009–2010 (60.2%: influenza A1N1 pandemic) to 2014–2015 (45.7%), after which it stagnated until 2019–2020 and rebounded in 2020–2021 during the COVID-19 pandemic (55.8%) [7].

The increase in vaccination uptake in the RCT could have been due to a Hawthorne effect (HE). A systematic review of an updated definition of the HE and its determining factors concluded that an HE was unlikely to explain an increase of more than 5% of vaccination uptake (article in submission).

Another reason for the increase may be a cohort effect. This cohort effect may be driven by a growing awareness triggered by the educational influence of patients’ GPs or other sources of public health promotion [8]. A cohort effect may also be driven by a perceived decline in health due to aging [9].

The main objective of this study was to assess the possibility of an HE, through comparison of the vaccination rate in the control group of the RCT to a third group of patients enlisted with GPs who were not aware of the RCT at the time the study was conducted. The secondary objective was to access the possibility of a cohort effect in this particular RCT. To investigate this, we followed our three cohorts during the two following years. As the seasonal influenza vaccination uptake was more or less stable, according to a time series analysis of the whole targeted population based on public health data, it was of interest to compare this trend with the trend in our cohort managed in primary health care [7].

## 2. Materials and Methods

We conducted a reanalysis of a cluster randomized RCT, with two arms, adding a third arm conforming to a Zelen design with retrospectively collected data. Data were extracted in 2019 from a routinely collected claim database at the patient level from 2014 to 2017. Clusters were constituted by patients aged 16 years or over registered with one of the participating 175 GPs from the Lille-Douai Health Insurance district (northern France) totalizing 155,025 patients.

The initial trial was a single-blinded 2/1 registry-based RCT carried out in the same area in 2014–2015 [2]. From the 75 GPs recruited in the trial, 25 were allocated to the intervention group and 50 to the control group using a computerized random draw. The design to achieve the first objective of the current study was described by Marvin Zelen in 1979 [10,11]. For this reason, we chose to call the third group of patients with retrospectively collected data the “Zelen group”. The recruitment of the 100 GPs of the Zelen group was conducted in 2019 and followed the order of a randomized list of GPs not approached during the recruitment for the original trial from the same insurance district. This Zelen-designed trial was thus a 4/2/1 registry-based cluster randomized trial, with the Zelen group not being aware or contaminated by the trial’s intervention, avoiding all risk of experimental artefact. To assess a potential cohort effect, the three groups were followed up during the two subsequent vaccination campaigns (2015–2016 and 2016–2017).

GPs had to practice actively in private practices, which could be single-handedly, in group practices, or in primary care interprofessional units. Primary health care was their main activity, excluding those where complementary medicine (e.g., homeopathy, acupuncture) or other practice (e.g., sonography, aesthetic medicine, angiology…) was their principal activity. Group practices from which one or more GPs had been involved in the original trial were excluded from enrolment in the Zelen group.

In the original trial, data were collected from the SIAM-ERASME claim database of the main mandatory health insurance fund covering about 75% of the insured population. For this reanalysis, data were collected from the “*Système National d’Information Inter Régimes de l’Assurance Maladie—Entrepôt”* (SNIIRAM-Warehouse) claim database, merging irreversibly anonymized data at the insurance beneficiary level from all French mandatory health insurance regimes. This larger database enhances the external validity of our results, as it includes certain professional categories like teachers or farmers, which were not included in the first trial database.

To be included in the study, participants had to be 16 years or over and registered by their health insurance regime in the Lille-Douai Health Insurance district with one of the 175 participating GPs. Registration on a GP’s patient list is not mandatory before the age of 16. For this reason, data regarding children are not reliable in French health insurance claim databases. Clusters were defined as all patients registered with a single GP.

The target population was defined as the patients who benefit from free access to the seasonal influenza vaccination, including patients aged 65 years or over and patients with a chronic condition requiring influenza vaccination coverage. Patients were informed about the anonymous use of their data and could refuse to participate. As research classified MR-004 by the French authorities, no informed consent for the use of individual data was required for each anonymized subject as data were routinely collected before the implementation of the European General Data Protection Regulation (GDPR), and all data were irrevocably anonymous (Ethics Committee of the Lille University Hospital (CPP Nord Ouest IV, advice #: HP 14/51) and the National Electronic Data and Liberty Commission (CNIL, advice 2019513)). The study was registered on ClinicalTrials.gov (registration #: NCT03239795).

The cohort was followed for three years after baseline. Patients without a chronic condition reaching the age of 65 during the four years of the follow-up and patients with a first chronic condition occurring during this period were not included. Patients who died or were lost to follow-up were not excluded.

The intervention consisted of withdrawing all informative material from GP waiting rooms (apart from mandatory information such as service fees), exposing only the 2014–2015 official Health Insurance poster promoting seasonal influenza vaccination and 135 official pamphlets. In the control group, the waiting rooms were left in their usual state. GPs from both groups knew that the seasonal influenza vaccination uptake of their patients would be measured, but GPs from the control group were not aware of the intervention. As written above, the 100 GPs from the Zelen group were unaware of the trial or the intervention. Their outcomes for the two-year time span of the trial were collected retrospectively.

It was not possible to measure the actual number of injections of seasonal influenza vaccination fulfilled by GPs and other medical specialists, self-employed nurses, midwifes, or community pharmacists. For this reason, the usual surrogate endpoint to measure influenza vaccination coverage in France was used: the main outcome was the number of vaccination units dispensed by community pharmacies for which payment appears in the claim databases of the health insurance compagnies under the name of the insurance beneficiary.

Data on patients encompassed their gender and age, the occurrence of a chronic condition, the date of deliverance of the vaccination unit, and their mandatory insurance fund. Data on GPs encompassed their gender and age, the number of patients on their patient list by year of interest, the number of patients aged 65 years and over, and the number of patients with a chronic condition of interest. In instances where patients lost their free vaccination voucher or when patients were not registered with a chronic condition of interest by the health insurance company, their GPs could prescribe a vaccination unit. The number of vaccination units prescribed by the GPs were taken into account for the published trial but appeared to be negligible. In this study, vaccination units prescribed by GPs were not considered.

The variables of interest to be extracted from the SNIIRAM-Warehouse database were transmitted to the data management center of the information processing department of the National Health Insurance Fund. A first extraction was unexploitable as no link was made possible between the anonymized patient lists and their referring GPs. A second extraction assigned a GP to each patient. Despite repeated requests, the data management center did not communicate the query algorithm in digital language as recommended by the CONSORT ROUTINE guidance [5].

A potential clustering of the outcome for patients treated by the same GP was considered. To correct this bias in computing the number of GPs needed for the trial with binary outcomes, an intracluster correlation coefficient of 0.02 was used, for α = 0.05 and β = 0.20 [12]. To find a difference of 5% between groups with a target size of 400 patients per GP, 75 GPs had to be enrolled (50 controls and 25 in the intervention group) [13]. As the maximal expectable HE was 5%, 100 GPs had to be enrolled in the Zelen group to be compared to the 50 in the control group.

The 75 GPs included in the original trial were enrolled between July 2014 and September 2015 following the order of a computerized random draw of 810 private practitioners registered as GPs by the health insurance fund. Another computerized random draw was used to allocate GPs to each group: 25 in the intervention group and 50 in the control group. To include the 100 GPs for the Zelen group, the continuation of the randomized list of the GPs from the Lille-Douai insurance district who had not been not approached when enrolling GPs in the original trial was used. Telephone calls were made between May and October 2017 to recruit these GPs, verify their eligibility, and obtain their agreement for participation, following the order of the list. Written consent of the eligible GPs was mandatory for inclusion.

Both the random allocation sequences were generated by the Public Health Department of Lille University Hospital. The participants in the original trial were enrolled by A.W.B. and S.Z.-M.M., co-authors of the study. The 100 participants in the Zelen group were enrolled by M.B.

Baseline characteristics of patients in the three groups are presented using a univariate analysis. Quantitative variables are expressed as a mean with a 95% confidence interval (95% CI) of the mean. Categorical variables are expressed as percentages and 95% CI of the percentage. The clustering variable “GP” was taken into account through the “*svydesign*” function of the package “*survey*” in R [14].

To assess the association between the vaccination status (dependent variable) and group (intervention/control/Zelen) membership, a generalized estimating equation (GEE) Poisson regression with an exchangeable working correlation matrix was used, resulting in risk ratios [15]. We adjusted for sex, age, having a chronic condition, and clustering by GP at baseline. To interpret the intervention effect, an interaction effect between time and intervention was included. Based on the assumed nature of the effect, we deemed the use of an interaction effect more accurate than the analysis of covariance (ANCOVA), which was used in the previous analysis of the trial [2,16,17]. To analyze the effect of being vaccinated the previous year on being vaccinated the consecutive year, we used a stationary first-order autoregressive transition model with being vaccinated the previous year included as a covariate [18]. To analyze the differences in the evolution of vaccination over time among age groups, point estimates and related confidence intervals, corrected for clustering by GP, were calculated and displayed after stratification per every five years of age. Analyses were carried out in R using packages “*geepack*”, “*survey*”, and “*ggplot2*” [14,19,20].

So far as possible, the reporting of this trial was implemented in accordance with the CONSORT extension for the reporting of randomized controlled trials conducted using cohorts and routinely collected data (CONSORT-ROUTINE) [5].

## 3. Results

The data at baseline were collected from 155,025 patients, and 23,024 patients were included in the analysis. Patients include those aged < 65 and with a chronic condition (*n* = 6354), patients aged ≥ 65 with a chronic condition (*n* = 10,961), and aged ≥ 65 without a chronic condition (*n* = 5709). The intervention group consisted of 3430 patients, the control group of 6620, and the Zelen group of 12,974.

The three groups did not differ at baseline in terms of age, gender, proportion of those aged ≥ 65 years, or the existence of a chronic condition (Table 1).

### 3.1. Main Outcome

The vaccination uptake in the three groups did not differ at baseline and after intervention (Table 2).

Comparing the three groups in a multivariable model, no difference was found between the control group and the intervention group despite a larger database and a more accurate analysis, supporting our previous findings (1). No difference was found between the control group and the Zelen group, acknowledging our hypothesis of a very weak or inexistant HE in our study. Comparing the intervention group and the Zelen group, we found a statistically significant difference (Table 3). However, this difference needs to be interpreted with caution as the baseline point estimate of the Zelen group was higher than the baseline point estimate of the intervention group; this may have elicited a regression to the mean (RTM) effect [20]. We therefore decided to perform a sensitivity analysis using analysis of covariance, which is known to adjust for a potential RTM effect [20]. This resulted in a nonsignificant intervention effect: RR 1.019 (95% confidence interval 0.986; 1.052).

### 3.2. Secondary Outcomes

Each subsequent year, vaccination uptake increased. Considering the point estimates, the increase was higher between the first year of follow-up (2015–2016) and the second year of follow-up (2016–2017) (Table 2 and Figure 1).

Over the whole period, patients vaccinated in the preceding year showed a 250% increase in the odds of being vaccinated in the subsequent year, or the odds of someone being vaccinated who was vaccinated in the previous year were 3.5 times the odds of someone who was not (Table 4).

Since age showed a significant effect on the evolution of vaccination over time (see Table 4), we assessed yearly vaccination uptake for the different age strata and per risk category.

The three groups at risk showed an increasing trend in terms of the percentage vaccinated at baseline per increasing age groups (see Figure 2, Figure 3 and Figure 4 and Table 5, Table 6 and Table 7). Within most groups, we saw an increasing trend in the percentage vaccinated during the three years of the study. For the group above 65 years of age, this increasing trend became less prominent for the older strata to finally reverse for the oldest stratum. For the group under 65, the increase within the group remained more stable over the different age strata. The percentage vaccinated at baseline in this group was relatively low, especially among the younger strata.

At baseline, the subjects with the highest vaccination uptake were patients ≥ 65 years with a chronic condition, with a mean percentage of 51.9% in the 65–69-year-olds, rising gradually to 70.3% in the 94–105-year-olds. However, the higher the value at baseline, the less it increased, reaching 75.6% to 71.0% in 2017 for the 65–79-year-olds and decreasing in the ≥85-year-olds. The major increase of 23.7% was observed in the most represented age group of 65–69-year-olds. (Table 5 and Figure 2)

In the group of patients ≥ 65 years of age without a chronic condition, the baseline percentage of vaccination uptake was lower, rising from 46.3% in the 65–70-year-olds to 69.2% in the 95–105-year-olds. However, this group showed the greatest increase reaching levels between 75.7% and 79.5% in the 65–84-year-old strata. A major increase of +29.4% was observed in the 65–69-year-old stratum. A decrease in the ≥90 strata was also observed (Table 6 and Figure 3).

In the group of patients < 65 years of age with a chronic condition, the mean baseline vaccination uptake percentages were rather low, with a baseline figure in the 16–39-year-olds of between 26% and 34%, and in the 40–64-year-old strata between 40% and 46%. The increase was gradual from 9.7% in the 16–24-year-old stratum to 35.7% in the 60–64-year-old stratum, reaching a main vaccination uptake of 74.9% in the latter (Table 7 and Figure 4).

In the end, the largest age group of 59–69-year-olds (*n* = 6729; 29.2%) in our population was also the group where the vaccination uptake increase was the highest. On the other hand, the vaccination uptake was diminishing or stable in the smaller group of patients (*n* = 2364; 10.3%) with a chronic condition and who were over 85 years of age, or without a chronic disease and over 90 years of age.

## 4. Discussion

Regarding the main outcome, no difference was found in the seasonal influenza vaccination units dispensed in community pharmacies between the control group and the Zelen group at baseline (winter 2013–2014) and after the intervention (winter 2014–2015). This rules out the hypothesis of an HE to explain the 4% to 5% increase in the vaccination uptake in the two groups of the RCT, while the uptake was diminishing by 2.5% in public health data on a time series analysis in the whole targeted population [7]. Initially, a difference was found between the intervention group and the Zelen group, but this difference disappeared after adjusting for RTM effect.

Our secondary objective was to explore a possible cohort effect to explain the increase in seasonal vaccination uptake in the study groups. Over the total three years of follow-up, we computed an increase of 19.0% from 51.4% to 70.4% in this cohort, nearly reaching the vaccination coverage of 75% recommended by the World Health Organization [21]. Vaccination uptake was strongly associated with being vaccinated the previous year in line with findings from other studies [22]. The increase in vaccination was determined by age, not by gender. Analyzing the three risk groups separately showed that the most substantial increase was observed in the two largest groups: the 2287 patients aged 60 to 64 years with a chronic condition (+23.7%) and the 1770 patients aged 65 to 69 years without a chronic condition (+29.4%). This increase was balanced by a decrease in vaccination uptake in the oldest stratum that we explain by uncensored mortality.

The reanalysis of our RCT published in 2018 confirmed that hanging posters and making pamphlets available in the waiting rooms to promote seasonal influenza vaccination did not increase the vaccination uptake between the two groups of the trial. This supports the evidence also noted by Li that posters and pamphlets can enhance patient knowledge, but they have limited educational impact to change patient behavior compared to encounters with doctors or nurses [23].

Our study has limitations. In our cohort, data from deceased patients were not censored, but assimilated to unvaccinated ones. This is our best explanation of a decrease in vaccination uptake in patients ≥ 95 years of age without a chronic condition or ≥ 90 years with a chronic condition.

The query algorithm in digital language to form our database from the SNIIRAM-Warehouse was not communicated by the health insurance company. For this reason, we are not able to discuss limitations that may have appeared during this process. However, the SNIIRAM is a powerful tool, a rather trusted claim database used by French public health authorities, encompassing almost the whole French population, and the engineers running the database are also experienced in running such data queries.

In the control and Zelen groups, no intervention in the waiting room was implemented. However, it is possible that posters or pamphlets from the influenza campaign might have appeared as our trial was implemented in real-life conditions. Based on our expertise and experience, we assumed that the presence of such promotional material would have been limited. Many public health campaigns are simultaneously implemented by posters and pamphlets in GP waiting rooms and are displayed for a long time, limiting the visibility of each of them. Maskell counted on average 72 posters covering 23 topics and 53 leaflets covering 24 topics with many outdated and poorly presented materials of limited accessibility [24]. For this reason, in the intervention group, we only displayed the material of the influenza campaign, withdrawing the material of all the others to enhance exposure.

To determine the seasonal influenza vaccination uptake, like public health authorities do and like we did in the original trial, we used a surrogate endpoint: the number of vaccination units delivered in community pharmacies [13,25]. It is uncertain whether all the delivered units were dispensed, as it is not known if patients had their vaccination units delivered by another path other than community pharmacies. However, the same limitation is applicable to all three arms of our study and should be without consequence for the final comparisons.

Our cohort only included persons who are registered with a GP. It is known that patients not registered with a GP in France have a lower vaccination coverage than registered ones [25]. As many GPs from the baby boom generation are going into retirement without a replacement, a growing number of patients (5.4 million in 2019), including older patients with a chronic condition (600,000 in 2019) have not found a new GP agreeing to register them on their patient lists [26]. Our findings may not apply to these patients.

As noted above, in this fixed cohort of patients managed by a GP, the vaccination uptake for seasonal influenza globally increased by 19% over three years. This increase, not noted in public health data from a time series analysis of the targeted population [7], can be characterized as follows: Firstly, the most prominent increase in seasonal influenza vaccination uptake happened among sexagenarians, between 60 and 69 with a chronic condition and between 65 and 69 without a chronic condition, representing one third of the total population targeted to be vaccinated. Secondly, the increase in vaccination uptake was generally lower in the younger age strata diagnosed with a chronic condition (diabetes, COPD…). Younger patients with a chronic condition diagnosed during the three years of follow-up were not included in the cohort, and hence their coverage rate did not influence our results.

It is possible that the remarkable rise in most of these age strata can be partly attributed to the influence of the GPs or other primary care staff. Based on a cross-sectional online questionnaire, Dexter identified seven independent factors that may result in an increase in vaccination uptake up to 7% [8]. However, these factors were analyzed in the context of the Quality and Outcomes Framework implemented in England, and hence are not fully transferable to France (there are no written reports to review influenza vaccine uptake rates in French primary care structures to earn quality rewards). Having a lead member of staff for planning the practice’s influenza vaccination campaign was identified as a key factor to promote vaccination in patients. In French surveys, the influenza vaccination uptake appears to be low among healthcare personnel in hospitals: the level is highest in personnel working in geriatric wards with 31% among nurses and 48% among physicians) [27]. In contrast, 77% of GPs declare to be vaccinated, 78% are promoting vaccination with their patients, and 93% are mentioning completed vaccinations in their medical record [28]. GPs in France may therefore function as an adequate lead member to plan influenza vaccinations. Further, Dexter indicates that sending a personal invitation to all eligible patients has a significant effect. All French patients targeted to be vaccinated receive such an invitation from their mandatory insurance company between the second half of September and the first half of October. This invitation favors the conversation about influenza vaccination initiated by the patient, by the GP, or by the community pharmacist who delivers the vaccination units [23]. Once convinced, patients usually find little barrier to access vaccination as they can receive it from their GP, midwife, community pharmacist, or nurse. This finding also elucidates the shortfall of vaccination uptake in populations not registered by a GP though noticed in cross-sectional studies [25].

It is of interest to note that the vaccination uptake increases in older adults though immune responses generally decline with age. Consequently a decline in vaccination efficacy can be expected, but data from RCTs searching influenza vaccine efficacy in older adults are contradictory [29]. The increase in vaccination uptake can be explained by the free vaccination of persons with a chronic condition and aged 65 years and over and by the incitation by authorities. It can also be explained by a growing perception of vulnerability in elderly patients who also may experience serious influenza infections with prolonged and sometimes incomplete recovery in themselves or in relatives [30]. This hypothesis is supported by the earlier increase in vaccination uptake in the population with a higher level of frailty related to a chronic condition. Separating efficacy and effectiveness of vaccination in persons ≥ 60 [31], GPs contribute to fostering influenza vaccination in elderly patients to prevent hospital admission due to influenza, to reduce primary care encounters for influenza-like illnesses, and to reduce influenza imputable mortality. However, observational studies considering all the confounding factors when evaluating vaccination effectiveness (with the most important confounding factor being the matching of vaccines to the circulating strains) remain contradictory [29].

## 5. Conclusions

The Hawthorne effect does not explain the rise in vaccination uptake in the two groups of the original RCT. Posters and pamphlets promoting seasonal influenza vaccination in GP waiting rooms have no influence on seasonal influenza vaccination uptake. In contrast, among patients registered with a GP, we saw an increase each year in follow-up in most of the age strata and more among the sexagenarians. This increase may be attributed to health promotion by primary care, nearly reaching the vaccination coverage recommended by the WHO.

## Figures and Tables

**Figure 1 vaccines-10-00826-f001:**
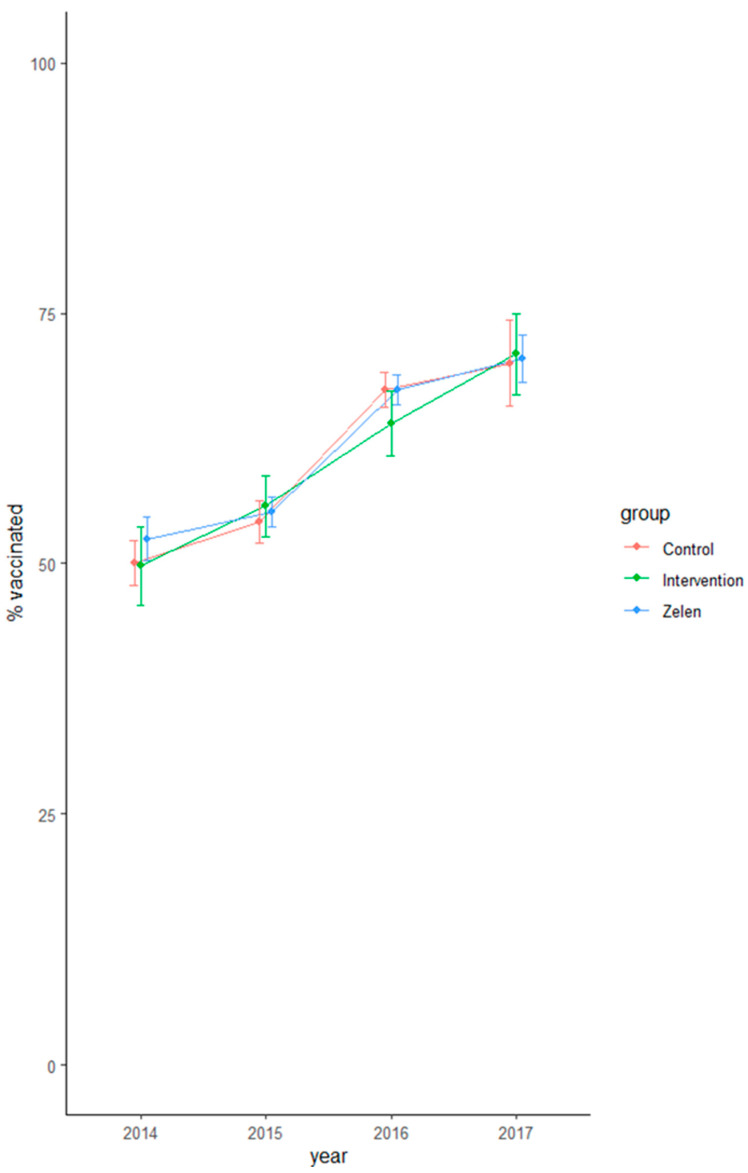
Vaccination uptake in the three groups. Legend: Estimate and 95% CI of the percent vaccinated per year and different study arms.

**Figure 2 vaccines-10-00826-f002:**
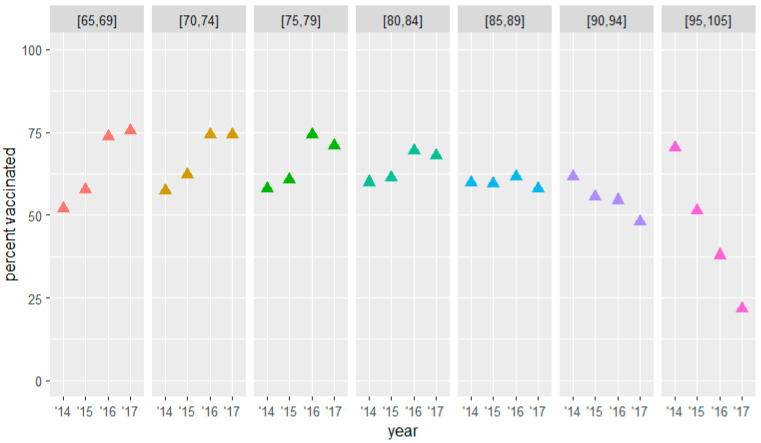
Influenza vaccination uptake by age at baseline ≥ 65 group (with chronic condition).

**Figure 3 vaccines-10-00826-f003:**
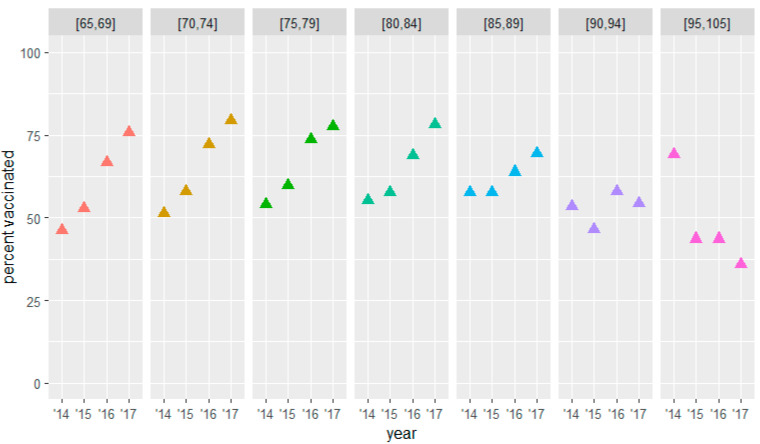
Influenza vaccination uptake by age at baseline ≥ 65 group (without chronic condition).

**Figure 4 vaccines-10-00826-f004:**
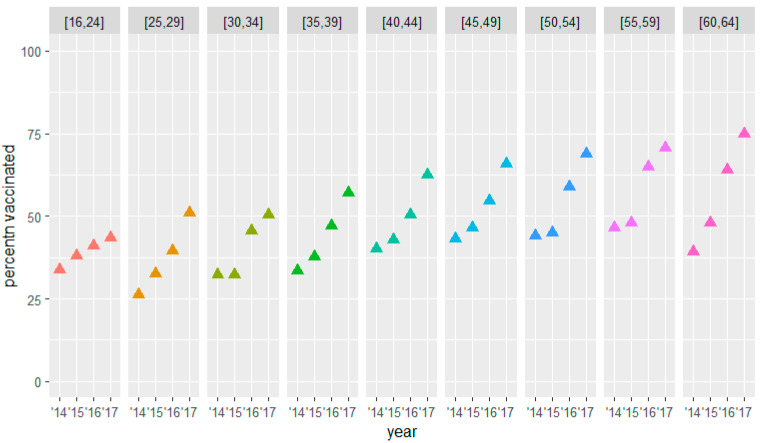
Influenza vaccination uptake by age at baseline < 65 group (with chronic condition).

**Table 1 vaccines-10-00826-t001:** Baseline characteristics.

Characteristics	Category	Intervention Group	Control Group	Zelen Group	P (i-c) adj	P (z-c) adj	P (z-i) adj
		(*n* = 3430)	(*n* = 6620)	(*n* = 12,974)			
		mean or % [95% CI]	mean or % [95% CI]	mean or % [95% CI]			
Age	years	70.3 [68.7; 71.9]	69.9 [68.9; 70.8]	69.7 [68.9; 70.5]	0.687	0.751	0.524
Gender	male	44.9% [42.2; 47.5]	43.7% [42.2; 45.3]	44.7% [43.4; 45.9]	0.465	0.353	0.889
Age ≥ 65	yes	76.0% [71.9; 80.0]	76.0% [73.5; 78.6]	75.1% [72.7; 77.4]	0.979	0.58	0.704
Chronic condition	yes	74.0 [70.9; 77.1]	74.7 [72.7; 76.8]	75.8 [74.3; 77.2]	0.700	0.413	0.314

Legend: CI = Confidence Interval, adjusted for clustering by general practitioner.

**Table 2 vaccines-10-00826-t002:** Influenza vaccination uptake in the three groups during three years after baseline.

Category	Intervention	Control	Zelen	P (i-c)	P (z-c)	P (z-i)
	(*n* = 3430)	(*n* = 6620)	(*n* = 12,974)			
	% [95% CI]	% [95% CI]	% [95% CI]			
2013–2014 (baseline)	49.8% [45.9; 53.7]	50.0% [47.8; 52.3]	52.5% [50.3; 54.7]	0.904	0.131	0.241
2014–2015 (intervention)	55.7% [52.6; 58.8]	54.1% [51.0; 56.3]	55.1% [53.6; 56.7]	0.408	0.444	0.748
2015–2016	64.0% [60.7; 67.2]	67.3% [65.6; 69.1]	67.4% [65.8; 68.9]	0.078	0.989	0.067
2016–2017	70.9% [66.9; 74.9]	70.0% [65.7; 74.3]	70.5% [68.1; 72.9]	0.763	0.851	0.856

Legend: CI = Confidence Interval, adjusted for clustering by general practitioner.

**Table 3 vaccines-10-00826-t003:** Comparison of the three groups after intervention.

Comparison	Estimate (RR)	95% CI	*p*
Intervention vs. Zelen	1.065	[1.020; 1.113]	**0.0043**
Intervention vs. control	1.037	[0.988; 1.087]	0.141
Control vs. Zelen	1.029	[0.995; 1.064]	0.09943

Legend: RR = Relative Risk; CI = Confidence Interval.

**Table 4 vaccines-10-00826-t004:** Factors associated with increased vaccination.

Characteristic	Estimate (OR)	95% CI	*p*
Vaccination in the previous year	3.50	[3.28; 3.73]	<0.001
**Characteristic**	**<65 (with chronic condition)**	**≥** **65 (without chronic condition)**	**≥** **65 (with chronic** **condition)**
	RR [95% CI]	RR [95% CI]	RR [95% CI]
Age	1.002 [1.001; 1.004]	0.996 [0.994; 0.997]	0.993 [0.992; 0.994]
Gender	1.021 [0.998; 1.045]	0.948 [0.928; 0.970]	0.990 [0.976; 1.004]

**Table 5 vaccines-10-00826-t005:** Influenza vaccination uptake by age at baseline ≥ 65 group (with chronic condition).

Age Category (Baseline)	*n* (Baseline)	2013–2014 (Baseline)	2014–2015	2015–2016	2016–2017
		% [95% CI]	% [95% CI]	% [95% CI]	% [95% CI]
[65, 69]	2672	51.9 [49.5; 54.4]	57.7 [55.6; 59.8]	73.7 [71.7; 75.7]	75.6 [72.5; 78.7]
[70, 74]	1922	57.3 [54.6; 60.1]	62.3 [59.7; 64.9]	74.3 [72.2; 76.5]	74.3 [71.5; 77.1]
[75, 79]	2266	58.1 [55.6; 60.5]	60.6 [58.2; 63.0]	74.2 [72.1; 76.3]	71.0 [68.5; 73.5]
[80, 84]	2045	59.9 [57.2; 62.6]	61.2 [58.8; 63.7]	69.4 [67.0; 71.8]	67.8 [65.4; 70.3]
[85, 89]	1378	59.7 [56.5; 62.9]	59.6 [56.4; 62.8]	61.6 [58.8; 64.5]	57.8 [54.7; 61.0]
[90, 94]	604	61.6 [57.7; 65.5]	55.6 [51.2; 60.0]	54.5 [49.7; 59.2]	47.8 [43.5; 52.2]
[95, 105]	74	70.3 [58.9; 81.6]	51.4 [41.6; 61.1]	37.8 [27.6; 48.0]	21.6 [11.9; 31.3]

Legend: CI = Confidence Interval, adjusted for clustering by general practitioner.

**Table 6 vaccines-10-00826-t006:** Influenza vaccination uptake by age at baseline ≥ 65 group (without chronic condition).

Age Category (Baseline)	*n* (Baseline)	2013–2014 (Baseline)	2014–2015	2015–2016	2016–2017
		% [95% CI]	% [95% CI]	% [95% CI]	% [95% CI]
[65, 69]	1770	46.3 [43.4; 49.1]	52.7 [50.3; 55.1]	66.7 [64.2; 69.2]	75.7 [73.2; 78.2]
[70, 74]	1092	51.2 [47.7; 54.7]	58.1 [55.0; 61.1]	72.1 [69.0; 75.2]	79.5 [76.0; 83.0]
[75, 79]	1156	54.0 [50.7; 57.2]	59.7 [56.6; 62.8]	73.6 [70.8; 76.5]	77.7 [74.0; 81.3]
[80, 84]	880	55.1 [51.1; 59.1]	57.6 [54.3; 60.9]	68.8 [65.3; 72.2]	78.2 [75.0; 81.4]
[85, 89]	542	57.6 [53.0; 62.2]	57.7 [53.2; 62.3]	63.8 [59.8; 67.9]	69.4 [65.6; 73.1]
[90, 94]	230	53.5 [45.8; 61.1]	46.5 [41.0; 52.0]	57.8 [52.2; 63.4]	54.3 [47.1; 61.6]
[95, 105]	39	69.2 [54.6; 83.9]	43.6 [27.4; 59.8]	43.6 [29.2; 57.9]	35.9 [20.3; 51.5]

Legend: CI = Confidence Interval, adjusted for clustering by general practitioner.

**Table 7 vaccines-10-00826-t007:** Influenza vaccination uptake by age at baseline < 65 group (with chronic condition).

Age Category (Baseline)	*n* (Baseline)	2013–2014 (Baseline)	2014–2015	2015–2016	2016–2017
		% [95% CI]	% [95% CI]	% [95% CI]	% [95% CI]
[16, 24]	166	33.7 [26.2; 41.2]	38.0 [30.3; 45.6]	41.0 [33.2; 48.7]	43.4 [35.8; 51.0]
[25, 29]	175	26.3 [19.6; 33.0]	32.6 [25.7; 39.5]	39.4 [32.3; 46.6]	50.9 [43.4; 58.3]
[30, 34]	189	32.3 [26.0; 38.5]	32.3 [25.9; 38.7]	45.5 [37.7; 53.3]	50.3 [43.2; 57.3]
[35, 39]	270	33.3 [27.4; 39.2]	37.8 [31.6; 44.0]	47.0 [40.7; 53.4]	57.0 [50.2; 63.8]
[40, 44]	385	40.0 [34.4; 45.6]	42.9 [37.7; 48.0]	50.4 [45.6; 55.2	62.6 [57.1; 68.1
[45, 49]	595	43.2 [38.8; 47.6]	46.6 [42.5; 50.6]	54.6 [50.7; 58.5]	65.9 [62.2; 69.5]
[50, 54]	922	44.0 [40.7; 47.4]	45.0 [41.6; 48.4]	58.9 [55.8; 62.0]	68.9 [65.7; 72.1]
[55, 59]	1365	46.4 [43.1; 49.6]	47.8 [45.1; 50.6]	65.0 [62.2; 67.7]	70.6 [67.4; 73.8]
[60, 64]	2287	39.2 [36.9; 41.5]	47.9 [46.0; 49.8]	64.1 [61.8; 66.4]	74.9 [72.1; 77.7]

Legend: CI = Confidence Interval, adjusted for clustering by general practitioner.

## Data Availability

Not applicable.

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
