# Peer review of "Reanalysis of a Randomized Controlled Trial on Promoting Influenza Vaccination in General Practice Waiting Rooms: A Zelen Design"

_vaccines, 2022, doi:10.3390/vaccines10050826_

Round 1
Reviewer 1 Report
In the manuscript “Reanalysis of a randomized controlled trial on promoting influenza vaccination in general practice waiting rooms. A Zelen Design”, the authors described interesting topic of promoting influenza vaccination in general practices’ in waiting room in the Lille-Douai district (Northern France). There are not many countries that achieve the WHO-recommended annual flu vaccination rate for their populations, so research into what might improve them is important.
However, before being published, the manuscript needs substantial corrections. The manuscript also contains a number of uncertainties that must be clarified, if it is to be published.
Some my comments on the manuscript are described below:
- Abstract:
The title of the manuscript is “Reanalysis of a randomized controlled trial ..., but in the abstract, the authors do not write when the reanalysis was done. They write about the study 2014-2015, which was already described in another manuscript. This should be carefully explained and specified.2. In Introduction
- Introduction
The introduction is not an introduction at all… and the introduction is not correct to a new study (Reanalysis of a randomized controlled trial ...,).
In the introduction, the authors described the methodology of the previous study (1), omitting the presentation of information, which in my opinion, should be included in Introduction section: of what influenza vaccination looks like in France (e.g. who pays for the vaccine?), What was the morbidity of influenza, the mortality from influenza during this study (compared with the previous study period) in the area of the Lille- Douai (Northern France). Has it changed? How long is the flu vaccination last in this area of France? In some European countries, influenza vaccination is given 2-3 months before the morbidity of flu increases. Influenza vaccination level in France and how it looks compared to other countries, for example EU countries.
Therefore, if they describe the promotion of influenza vaccination in France, it should be described. What are the forms of influenza vaccination promotion in France and refer to examples in other countries.
In line 52 .... stagnated until 2019-2020, and rebounded in 2020-2021 during the covid-19 pandemic (55.8%) [4]. - But your study was not done during 2019- 2020-2021.
- Material and Methods
In Material and Methods, the authors cite their work five times (1). Not too many times ? This is intended to be a new manuscript, not the old rewritten manuscript.
Again authors write ” As described in the report of the original trial, the design was a single blinded 2/1 registry-based RCT carried out in the area of the Lille-Douai Health Insurance district (Northern France) in 2014-2015 [1]”.
Here, the authors should write the total number how many patients, GPs offices took part in this study. When was this study (Reanalysis RCT) conducted, what period was it covered, etc. Reverse the order of the test description and do not write about what has already been described (copy - in the previous manuscript 1). Please correct.
Line from 93 to 100 . The entire paragraph is not referenced. Please, add references.
Line 103 ….with one of the 175 participating GPs.
This is what I am writing about above. Why suddenly 175 GPs ?? By changing the order of the article description, it will be known that 175 GPs took part in the entire study, the first part was 75 GPs, and in the current one 100 GPs were added, using the method- A Zelen Design
Line 107 … - No informed consent for the use of their individual data was required for each anonymized subject as data were routinely collected before the implementation of the European General Data Protection Regulation (GDPR) and all data were irrevocably anonymous. The entire paragraph is not referenced. Please, add references.
Moreover, in my opinion, it is not correct: the European General Data Protection Regulation (GDPR) has nothing to do with research. The study collected data on insurance and patients' diseases from GPs and the consent of the Ethics Committee is always required.
Without the consent of patients to participate in the research, the research does not comply with the ethics of conducting research. This is a very serious objection.
Line 78-79 The recruitment of the 100 GPs of the Zelen group was conducted in 2019- And how do you know whether or not there were leaflets o influenza vaccination in 2015/16 and 2016/17 in general practice waiting rooms in those time. This Zelen group was formed among doctors who had not previously participated in the study. Please explain this.
Line 135-137 When patients have lost their free vaccination voucher or when patients have not been registered with a chronic condition of interest by the health insurance company, their GPs can prescribe a vaccination unit. - And here there is no explanation of the payment system or the lack of for influenza vaccination in France (this should be described in the introduction). Where does the patient get the free flu vaccination voucher? from GP? This is unclear to readers from other countries, and the manuscript is submitted to an international journal. Please explain this.
Line 141-146 Despite repeated requests, the data management center did not pass the query algorithm as recommended by the CONSORT ROUTINE guidance (2).- Why was it supposed to consent? Did the patients taking part in the study give their consent? Is there a consent of the Ethics Committee to obtain such data? Please explain that.
Line 161-162. Has it been approved by the Ethics Committee (EC)? Please add number of the EC.
Line 166. what do these abbreviations mean? Please explain them.
- Results
Line 198. In the manuscript (1), the control group counts 6,816, and intervention group is 3,781; which does not agree with the current data (Table 1) of these groups (3,430; 6620 respectively). The more so because the population has expanded to include people from 16 years of age. Additionally, they are minors, was there a parental consent for data collection? Is there approval from the Ethics Committee? Please explain it.
Table 7 age 16-24 but the same Figure 3 age 15-24
Why do you present the results only Influenza vaccination uptake by age at baseline <65 group with chronic condition, and omitting people <65 without chronic condition.
- Discussion
The discussion is not a discussion, but a repetition of describing the method and results.
Please refer your results to the work of other authors and do not repeat the description of the research method and results.
Line 276-293. This is not a discussion, but a third time describing the method and the results.
Line 294-297. This should be described in the results and not in the discussion again.
Line 299-305. again description of the results
Line 306-308 The seventh time citation of author's manuscript (1).
Line 309-311. In the middle of the discussion, which for now is not discussion but describing the results, the authors write about the limitations. The limitations are described throughout the discussion.
Line 319-324. this is not a discussion
Line 326-331. There is not references. Please, add references.
Line 332-341. There is not references. Please, add references.
Line 354-362. At the end of the discussion, we find out what flu vaccination looks like in France. This should be described in the introduction.
- Conclusion
Line 366-372. This is not a response to the title of the article, besides, the authors did not test vaccination among 6-year-olds, but among people over 16 years of age.
Author Response
Dear reviewer,
Thank you for the accuracy of your review. We guess that it contributed to the quality of the article, mainly regarding two supplementary paragraphs in the introduction to present briefly the seasonal influenza epidemics in France and the influenza vaccination politics of health authorities in France compared to Europe. It also contributed to clarify the ethical perspective of RCTs based on big data collected in routine: regulatory obligations when collecting the data (GDPR) and obligations to be allowed to use these data in research (GDPR and MR-004).
Please, see in attachment the point by point response to your relevant comments

Reviewer 2 Report
In this paper Berkhout and coll. reanalysed factors promoting influence vaccination in general practice, concluding that the observed increase of vaccination rate over a three-year time span was neither influenced by a Hawthorne effect nor by the presence of informative material in GPs waiting rooms.
Despite some bias, exhaustively commented by the authors, the paper underlines the importance of primary care in health promotion.
- The authors observed an increase in vaccination adherence by age, and a decrease in the oldest stratum, irrespective to the presence of a chronic condition.
Could this fact be influenced by the awareness to belong to a “frail” population, simply due to age?
- Among factors influencing vaccination, the family endorsement and the presence in the family of people with chronic conditions could explain the increase of vaccination in the different strata?
- In the oldest stratum the presence of a “family” or not could explain the decrease of vaccination?
Author Response
Dear reviewer,
Thank you for the interesting and relevant points that you raised. The patients in the SNIIRAM Warehouse database being totally anonymized, it is quite impossible to have family related data. However we added a paragraph to the discussion raising another reason than the management by the GP to explain the raise of vaccination uptake in sexagenarians.
Please, see in attachment the point by point response to your comments

Round 2
Reviewer 1 Report
The authors of the manuscript "Reanalysis of a randomized controlled trial on promoting influenza vaccination in general practice waiting rooms. A Zelen Design” have already made major corrections and responded to the reviewer's opinion.
However, the discussion is not fully corrected.
It was asked not to re-describe the methodology of the study in Discussion section, but this still dominates the discussion.
Line 867-876 and Line 877-954 and again 955-957 and 962-967 and etc.
In fact, these paragraphs do not add anything new, and they do not even compare to the works of other authors from other countries.
In the sentence “Many public health campaigns are simultaneously implemented by posters and pamphlets in waiting rooms, and are displayed for a long time, limiting the visibility of each of them [6]”, the authors cite only their work. Are only the authors doing research on flu vaccination campaigns?
A large number of the 28 positions in the publication are older than 10 years or authors focus on citing their own publications and other French authors. In such situation the manuscript loses its publication value in an international journal because it only discusses the issue in the local aspect and is suitable for publication in the regional journal. It is therefore necessary to improve the discussion section.
Author Response
Thank you for the fast review of our manuscript you provided.
Since a short reminder of the methodology was not necessary to the general understanding of the text or the limits of the study, it has been removed from the discussion.
I agree that citing only our work might appear somewhat pretentious. However, the cited article is a review of 14 articles published in the field of our research from 2004 to 2017, frequently cited by other authors. We found 6 more recent articles in line with our work of which 3 were cited
After having added three more references to reach 31 in total, we cited
- two articles we published on this topic in English, one of these being the article of the original RCT and the other one being a review of 14 article in the field of this research.
- Six articles contribute to describe the influenza epidemic and vaccination procedures in France as you rightly suggested we should do
- Two European article situating the French background compared to the rest of Europe, as you rightly suggested
- Twelve methodology articles (design, analysis and report) to sustain methodological choices
- Nine articles from other European and non-European countries, comparing our findings to others in the field (six plus the 3 that were added)
We regrettably found few recent articles in the field of vaccination promotion in GPs’ or primary care waiting rooms. However, we tried to keep our reference list not excessively long, and we tried to balance it.
Please, see the attachment
